# ECONOMICS ARENA FOR LARGE LANGUAGE MODELS

## ABSTRACT

Large language models (LLMs) have been extensively used as the backbones for general-purpose agents, and some economics literature suggest that LLMs are capable of playing various types of economics games. Following these works, to overcome the limitation of evaluating LLMs using static benchmarks, we propose to explore competitive games as an evaluation for LLMs. By varying the game history revealed to LLMs-based players, we find that most of LLMs are rational in the sense of playing strategies that can increase their payoffs, but not as rational as indicated by Nash Equilibria (NEs). Moreover, when game history are available, certain types of LLMs, such as GPT4, can converge faster to the NE strategies, which displays higher rationality level in comparison to other models. In the meantime, certain types of LLMs can win more often when game history are available, and we argue that the winning rate reflects the reasoning ability with respect to the strategies of other players. Throughout all our experiments, we observe that the ability to strictly follow the game rules described by natural languages also vary among the LLMs we tested. We provide an economics arena for the LLMs research community as a dynamic simulation to test the above-mentioned abilities of LLMs, i.e. rationality, strategic reasoning ability, and instruction-following capability.

## 1 INTRODUCTION

In the past decade, researchers have successfully applied deep learning (DL) models to various cognitive tasks (Silver et al., 2016; Krizhevsky et al., 2017; Vaswani et al., 2017), and obtained impressive results. Among these DL models, Transformer (Vaswani et al., 2017) based large language models (LLMs), such as GPT3.5 (Brown et al., 2020) and Llama2 (Touvron et al., 2023), have shown great potential in acting as general-purpose agents (Zhang et al., 2023; Wang et al., 2023c). Evaluating these models, however, is challenging. Many works have been proposed to test models' performance on either a large-scale static benchmark such as (MMLU, Hendrycks et al., 2021a), or with A/B tests judged by humans (Ganguli et al., 2023). One common and evident limitation of these methods, however, is that the environment for the model to be tested is *static* (Aiyappa et al., 2023; Zhou et al., 2023), thus cannot reflect whether the models can respond *adaptively* to a *dynamic* environment.

Given the capabilities of LLMs to complete tasks through text-based interactions, economists, such as Horton (2023); Phelps & Russell (2023), have applied LLMs to many typical economics games, either cooperative or competitive. Inspired by their works as well as the emergent price bargaining ability of LLMs showed by Fu et al. (2023), we propose to evaluate LLMs in simulated number-based competitive games, which makes it possible for us to put forward the quantitative metrics for measuring the performance of LLMs in the following aspects. To base our work on a solid assumption, we first verify whether the selected models can perform rationally in two types of competitive games, beauty contests and private-value second price auctions. Here, we follow the most common rationality assumption used in economics, where a rational agent should aim to maximise one's utility (Mas-Colell et al., 1995). we can then dynamicise the games by controlling both the configurations of the games, e.g. the price of items in auction games, and the selection of LLMs as the backbones of players, e.g. with GPT4 and GPT3.5 as bidders or with GPT4 and Claude2 as bidders. By tracking whether the strategy distribution of the agents is responsive to the environment changes, we can verify which LLMs can adapt to the dynamic environments. After the above verification about the models' rationality and adaptation ability, we can let the LLM-based agents play in

our economics arena, and track the change of their payoffs to reflect their average capability of understanding the game instructions written in natural languages and proposing strategies to increase utilities. Moreover, since we humans often win games through learning and reasoning about the strategies of other players (Scontras et al., 2018), we also propose to measure the strategic reasoning ability of LLMs by exposing them with varying amount of game histories. We hypothesise that LLMs with stronger strategic reasoning ability would win more often, as captured by the winning rate, and learn faster, as shown by the convergence rate to the optimal strategies.

To provide the community with a publicly available simulation package to evaluate LLMs, we design and implement an economics arena (`EconArena` for short) which consists of several types of auctions and beauty contest games Through our experiments with a series of LLMs, we show that:

1. When multiple agents back-boned by LLMs play the games in our economics arena, w/o game history information, only certain models can obtain positive payoffs, and they all fail to achieve the NEs, which suggests that are rational, but the rationality degree can be further improved;

2. Certain LLMs can adapt to game configurations as well as changes in their opponents' strategies, which suggests that these models are responsive to dynamic environments;

3. When game history is available, certain LLMs win more often than the others, which we argue that they show stronger strategic reasoning ability;

4. Similarly, certain LLMs can converge to the optimal NE strategies faster when competing with a rational baseline. By controlling the ability of opponents, this result reflects certain LLMs to hold a higher degree of rationality and in-context learning capacity compared to others.

Based on the experiments, we also found that some LLMs break rules more often than others as a by-product. Thus, we argue that the fraction of rule breaking over the total number of games played can also reflect the ability of LLMs in comprehending natural language instructions.

## 2    RELATED WORKS

**LLM-based Agents** Autonomous agents have long been recognised as a promising avenue for pursuing artificial general intelligence, expected to carry out tasks through self-directed planning and actions (Wang et al., 2023b). Recently, as LLMs have showcased remarkable emergent capabilities in executing various tasks (Brown et al., 2020; Ouyang et al., 2022; Wei et al., 2022; OpenAI, 2023), researchers have begun to utilise LLMs as the planning and reasoning engine for agents (Weng, 2023; Ahn et al., 2022; Park et al., 2023; Wang et al., 2023a; Yao et al., 2023b; Sumers et al., 2023; Liu et al., 2023b; Sumers et al., 2023). Backboned by LLMs, these agents can easily understand natural language and thus follow instructions. Moreover, with techniques such as Chain-of-Thought (CoT, Wei et al., 2023a; Yao et al., 2023a) and problem refinement (Xi et al., 2023), they can also exhibit reasoning and planning abilities. Furthermore, by learning from feedback and performing new actions, these agents are able to achieve interactive capabilities with the environments (Liu et al., 2023a; Shinn et al., 2023; Lin et al., 2023). The language modelling, understanding, and reasoning capacity offered by LLMs bring the LLM-based agents closer to human-like intelligence. In the meantime, these LLM-based agents have also been applied to various real-world scenarios, such as believable simulacra of human behaviour (Park et al., 2023), industrial automation (Xia et al., 2023; Ogundare et al., 2023), software development (Qian et al., 2023), and scientific research (Boiko et al., 2023).

**LLMs as Players in Games** Recently, there is an emerging line of research on the ability of LLMs to replicate human behaviour in game theory. Brookins & DeBacker (2023); Kasberger et al. (2023); Guo (2023); Phelps & Russell (2023); Akata et al. (2023); Gandhi et al. (2023); Horton (2023) explored the potential of using LLMs as players in strategic games, such as the Prisoner's Dilemma and the Dictator game. Guo (2023) found that `GPT3.5` exhibits some difficulty in performing strategic reasoning, while `GPT4` displayed higher strategic reasoning ability. Phelps & Russell (2023) illustrated that `GPT3.5` can, to some degree, incorporate the concepts of altruism and selfishness within the iterated Prisoner's Dilemma. However, the majority of simulated agents struggled to adjust their strategies adequately when confronted with varying degrees of cooperation or defection from their opponents. Brookins & DeBacker (2023) found that `GPT3.5` displays preferences for fairness and cooperation, often exceeding those elicited from humans in laboratory experiments.

Besides, Xu et al. (2023) explored LLM-based agents in Werewolf games, which is a representative and well-studied incomplete information game and found emerging social behaviours, such as trust, confrontation, camouflage, and leadership, from `GPT3.5`.

**Conventional Evaluation of LLMs** Understanding the essence of intelligence and establishing whether a machine embodies it poses a thought-provoking question for researchers. There have been a number of comprehensive benchmarks aiming at evaluating LLMs from different aspects, such as MMLU (Hendrycks et al., 2021a), BIG-bench (Srivastava et al., 2023), HELM (Liang et al., 2022), C-Eval (Huang et al., 2023), and AGIEval (Zhong et al., 2023). These benchmarks can be roughly divided into two categories: knowledge-oriented and reasoning-oriented. Knowledge-oriented benchmarks, like MMLU (Hendrycks et al., 2021a) and CEval (Huang et al., 2023), aim to evaluate the capacity of world knowledge. In contrast, reasoning-oriented benchmarks, like GSM8K (Cobbe et al., 2021), BBH (Suzgun et al., 2022) and MATH (Hendrycks et al., 2021b), focus on evaluating the capability of solving complex reasoning tasks. In addition to employing benchmarks for evaluating LLMs automatically, Chatbot Arena (Zheng et al., 2023) has developed a crowdsourcing platform that enables users to engage in conversations with two anonymous chat LLMs and report pairwise comparison results. Despite the current flourishing developments in the evaluation of LLMs, the existing evaluation methods still focus more static environments, and fall short in assessing strategic reasoning which is a crucial facet of human intelligence.

## 3 ECONOMICS ARENA

### 3.1 SINGLE-ROUND COMPETITIVE GAMES WITH IN/COMPLETE INFORMATION

At the moment, we provide only single-round games in `EconArena` , such that the LLMs can be used in an "off-the-shelf" fashion. In this way, we introduce the least amount of prompt engineering onto the performance of LLMs, and put all relevant game information in a single prompt to avoid potential influence from techniques like dialogue management Brabra et al. (2021). When running LLMs in `EconArena`, every single run of the games is only a single-round dialogue, and no multi-round dialogue is needed.

We focus on competitive games with unique pure NE in the initial version of our economics arena. Such games allow conflicts between agents, and winning strategy is not necessarily a strategy bringing positive payoffs. Having a unique NE ensures there exists an optimal strategy agents can adopt, and no one can obtain better outcome by deviating. To be more specific, we implemented the second-price auctions, and beauty contest games, in the alpha version of `EconArena`[1]. More formal descriptions of the games are provided in Appendix A.1.

Our auctions are private value, thus incomplete information, second price auctions where the player bidding the highest price gets the item but only need to pay the second-highest price, and the unique pure NE is for all players to bid their private value. Since the private value of the bidding item vary for bidders backboned by LLMs and the winner pays the second-highest price, the players have to reason about the others' strategies in order to maximise their own utilities. Moreover, the strategies to bid the items are not necessarily profitable strategies, since a player can get the item with a negative payoff through overbidding. During our experiments, we indeed find that certain LLMs indeed tend to overbid to get the item.

Unlike the auction games, beauty contest games are of complete information, and also have only one unique pure NE. However, the players still have to reason about others' strategies in the beauty contest games, as the winner is decided by the *average* of numbers from *all players* and players have to guess/approximate the numbers of others in order to win. All in all, we argue that our `EconArena` is a *multi*-agent environment, where the players have to reason about others' strategies in order to maximise their own payoffs.

### 3.2 METRICS OF INTEREST

We hereby propose and list some metrics arisen in our economics arena that are interesting for either LLM researchers or economics researchers.

---

[1]We will introduce more types of games in the updates of `EconArena`, not only competitive games, but the cooperative games.

- **Payoff changes over games**: the first and foremost metric of various research interest is the change of payoffs over the arena games, which is measured by $u_i^g$, i.e. the utility function from game $g$ to player $i$. Furthermore, since all arena games have only one unique NE, suppose the utility of player $i$ under game $g$ is $\bar{u}_i^g$, $u_i^g - \bar{u}_i^g$ can also be of interest as it can reflect how close the player's strategy $s_i^g$ is close to the NE which is the optimal strategy when all players are rational.

- **Strategies over game configurations**: strategies of the player $i$ in a game $g$ with varying configurations $c$, $s_i^g(c_j)$ where $j$ indexes different configurations, is also an interesting metric, as it can reflect whether $s_i^g(c_j)$ is *adaptive* or *consistent* over game configurations $c_j$ where $j \in \mathbb{Z}^+$. One example of adaptive strategies is the bidding price under different budgets in auction games, where a player should always bid a price less or equal to its budget. Thus, when the budgets and private values are varied, the strategies of the players should also change as a *response* to the changed configurations. Regarding the consistent strategies over configurations, an example is the range for guessing numbers in beauty contest games which can be $[0, 100]$, $[0, 1000]$, and $[0, 10000]$. Since the winning strategy is consistent across the ranges, ideally, whatever the range is, the winning strategy from a player $i$ should be *consistent* and *robust* to the varying range for guessing numbers.

- **Strategies over player configurations**: another interesting thing to observe is whether a player $i$ plays different strategies when the opponents in the games changed. When all other players in a game are rational, a more rational strategy should always be preferred, as irrational strategies definitely lead to lower payoffs. On the other hand, when the other players in a game are mostly irrational, a rational strategy may not be a winning strategy. For example, suppose there are 5 players in a beauty contest with range $[0, 10]$, and 4 of them propose 10 while the remaining one propose 0. Since the $\frac{2}{3}$ of the average in this case is $\frac{16}{3} > 5$, the most rational strategy 0 actually loses. Therefore, the strategies of a player under different play configurations can show whether an LLM is actually adapting to the dynamics brought by the other players, especially when game history is available, which is illustrated below.

- **Strategies over game history availability**: another importance factor for the performance of the LLM-based agents is the in-context learning capability (Wei et al., 2023b). To test whether and how well the players can improve their strategies through in-context learning, our economics arena can provide the game history for the previous runs in the same session. Through our experiments, we did find that the performance of the certain models, such as `Claude2`, hugely improve through utilising the history information for the previous runs in the same session.

Discussion about the above metrics in further details can be found in Appendix A.3.

## 4 EXPERIMENT RESULTS

To demonstrate that `EconArena` can indeed capture the interesting metrics introduced before, we take the following LLMs as players to cover a sufficient number of popular models: 1) `GPT4` from OpenAI (2023); 2) `GPT3.5` from OpenAI (2022); 3) `Claude2` from Anthropic (2023b); 4) `Claude-I` (Claude-Instant-1.2) from Anthropic (2023a); 5) `PaLM2` from Anil et al. (2023); 6) `Llama2` from Touvron et al. (2023); 7) `Baichuan2` from Yang et al. (2023); 8) `ChatGLM2` from Du et al. (2022); Zeng et al. (2022); 9) `ChatGLM3` from also Du et al. (2022); Zeng et al. (2022). Since we are not certain whether the responses to the given prompts are sampled out or greedily searched out from LLMs service provides, we run multiple independent runs of the games for the following experiments. Details about our prompts in different setups are given in Appendix B.

### 4.1 NON-MAXIMALLY RATIONAL BEHAVIOURS OF LLMS IN ECONOMICS ARENA

To verify the rationality assumption from economics, we first investigate whether the LLMs in `EconArena` are seeking to increase their payoffs under various scenarios. Results of 9 LLMs playing beauty contests and second-price auctions are shown in Figure 1 and 2 respectively. The results are from 150 independent sessions.

As shown in Figure 1a, the mean payoffs are almost 0 in the beauty contests for `ChatGLM2`, `ChatGLM3`, `Llama2` and `PaLM2`, but positive for the rest, and `GPT3.5` outperforms all. However, this comparison does not imply stronger rationality of `GPT3.5` among the LLMs straight

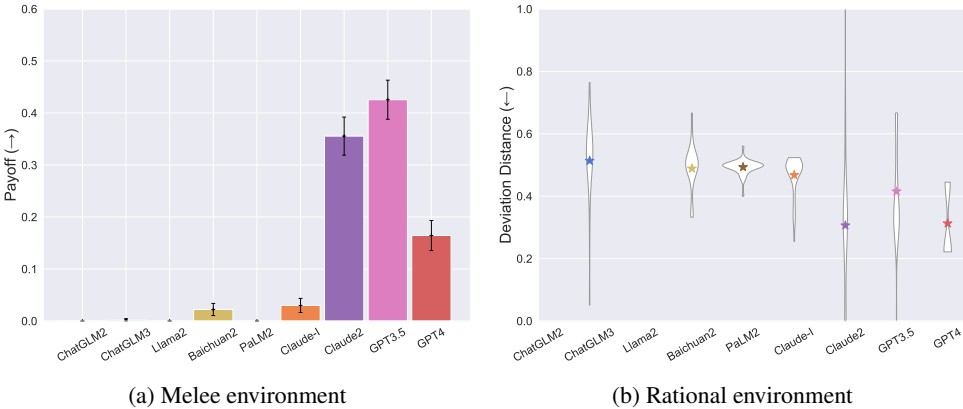

(a) Melee environment                    (b) Rational environment

Figure 1: The performance of LLMs in beauty contest games playing against different types of opponents. By "Melee environment", we refer to the case where the LLMs are playing against each other, while in the "Rational environment", the LLMs are playing against 4 hard-coded rational agents. Note that some results for ChatGLM2 and Llama2 are not recorded because they failed to complete the games.

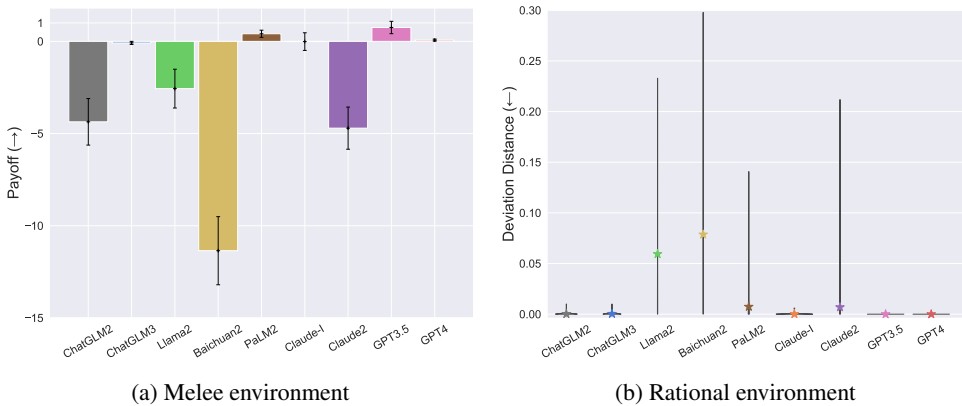

(a) Melee environment                    (b) Rational environment

Figure 2: The performance of LLMs in second price auction games playing against different types of opponents. By "Melee environment", we refer to the case where LLMs are playing against each other, while in the "Rational environment", the LLMs are playing against 4 hard-coded rational agents. Note that Figure 2b is a vilolin graph, same as Figure 1b.

away, as there are two potential causes behind the agents' behaviours: 1) the LLMs themselves are not maximally rational; 2) the LLMs believe that the opponents they are facing are not rational, thus they choose non-maximally rational strategies. To disentangle the reasoning behind this, we explore another set-up: each LLM is playing against 4 hard-coded rational agents, and they were each given explicit prompts that they are playing with rational opponents. Henceforth, this is denoted as the "rational environment", and serves as the baseline for all comparisons. Figure 1b displays that despite knowledge of rational opponents, all models still deviate away from NE, implying that they are not maximally rational. Out of the LLMs, Claude2 and GPT4 fare better by having lower average deviation distance, GPT3.5 is slightly worse than the two, but remains better than the rest. We can interpret this as Claude2 and GPT4 being more rational than the other LLMs.

For second price auctions in Figure 2a, ChatGLM3 receives the highest average payoffs and Baichuan2 obtains the lowest. Similarly, we cannot judge the rationality of the models solely based on this result. In Figure 2b, we again illustrate a "rational environment" and investigate the actual payoffs against NE payoffs via deviation distance. Most models can obtain payoffs very close to NE, except for Llama2 and Baichuan2 . PaLM2 and Claude-I 's deviation distances differ from 0 only slightly. By controlling for the rationality perception about other players, it is possible to infer that in the second-price auction, ChatGLM2 , ChatGLM3, Claude-I ,

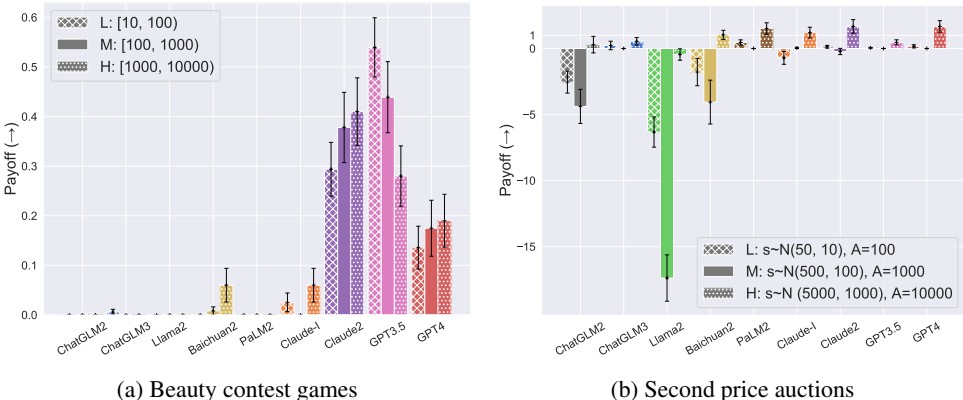

(a) Beauty contest games

(b) Second price auctions

Figure 3: Average payoffs (↑) of LLM when varying game configuration. In the beauty contest setup, we change the upper bound of the interval from which an agent chooses a number, while in the second price auctions, we vary the private value signals ($s$) and asset level ($A$).

GPT3.5 and GPT4 are behaving rationally on average. Once this restriction is stripped away (shown in Figure 2a), most of these models receive close to 0 or even negative payoffs, except for GPT3.5 . This implies that GPT3.5 not only has stronger or comparable rationality level compare to the other LLMs, it is also better at selecting rationalisable strategies when rationality of other players may not be guaranteed.

## 4.2 ADAPTION OF LLMs TO DYNAMIC ENVIRONMENTS

In order to analyse if LLMs are indeed responding and adapting their strategies to the dynamic environments, we vary the game configurations to dynamicise the environments, as well as vary the models to dynamicise the players.

**Varying game configuration:** In the beauty contests, this variation is achieved by changing the interval from which an agent chooses a number from. The experiments are divided into three groups (L, M, H), in each group the upper bound is randomly generated from a different range. For instance, L: $[10, 100]$, M: $[100, 1000]$, H: $[1000, 10000)$. In a sense, the potential strategy space expands across groups. 50 sessions, with 1 run per session, were conducted. In Figure 3a, we show that for all three groups, Claude2 and GPT3.5 are performing substantially better than the rest of the LLMs in achieving higher average payoffs, and GPT4 comes in third. The other LLMs have average payoffs that are relative low. For the strongest two LLM, varying the game configurations have different impact. Claude2 shows improvement in payoffs as upper bound increase, while GPT3.5 shows a decline. This not only suggests that having larger strategy space do affect LLM performance, it could also imply that as the ability to rationalize strategies decreases for all LLMs, there could be a "spillover" as GPT3.5 becomes substantially and adversely impacted by the increase in upper bound, thereby increasing the average payoffs for the other models.

For second price auctions, variations in private value signals $s$ and asset level ($A$) are implemented by dividing the experiments into three groups, where L: $s \sim N(50, 10)$, $A = 100$, M: $s \sim N(500, 100)$, $A = 1000$, H: $s \sim N(5000, 1000)$, $A = 10000$. Since private values influences the amount one is willing to spend on acquiring the item, thus to vary the private values, assets would need to vary correspondingly in the game configuration. Figure 3b shows that PaLM2 achieves the highest average payoffs in L, as private distribution varies and assets increase, PaLM2 , Claude2 and GPT4 obtain comparable payoffs that are better than the other LLMs in H. As for M, almost all LLMs earn either negative or close to 0 average payoffs. Llama2 perform poorly across all groups. The higher mean value of private signals, as well as the higher assets are both expected to lead to more aggressive bidding as agents would have more to lose if they did not manage to win the bid and they can also afford to spend more. However, the higher standard deviation in private signals also imply there is greater variability among bidders, thus higher level of uncertainty, which can lead to more cautious bidding. The combined effect of the above could have resulted in the average payoff pattern across the groups.

| LLMs | Beauty contest games | | | Second price auctions | | |
|---|---|---|---|---|---|---|
| | 20 Sessions | 60 Sessions | 100 Sessions | 20 Sessions | 60 Sessions | 100 Sessions |
| Baichuan2 | 0.050 | 0.033 | 0.020 | 0.865 | 0.473 | 0.284 |
| Claude-I | 0.000 | 0.000 | 0.030 | 1.000 | 1.190 | 1.062 |
| Claude2 | **0.450** | **0.392** | 0.370 | **1.454** | **1.200** | 0.991 |
| GPT3.5 | 0.400 | **0.392** | **0.410** | 1.012 | 0.870 | **1.206** |
| GPT4 | 0.100 | 0.183 | 0.170 | 0.734 | 1.063 | 1.084 |

Table 1: Average payoffs (↑) of LLMs in the "senior environment", where we hand-picked 5 LLMs to test the impact of variation in opponent types.

In both games, unlike previous conjecture about consistent strategies, LLMs do display adaptation to changes in the environment, where their strategies and the corresponding average payoffs varies with changes in game configurations.

**Varying player configurations:** To test the impact of changes in player configurations, particularly opponent types, we hand-picked 5 LLMs that perform better in the "melee environment" and created a "senior environment". Within which, they play the games for 20, 60 and 100 sessions.

For the beauty contest games, as shown in Table 1, Claude2 and GPT3.5 obtain the highest average payoffs among the selected models, with Claude2 does better than GPT3.5 for 20 sessions, and the reverse happens for 100 sessions. This implies that GPT3.5 is less variable in its behaviour than Claude2 and therefore has less volatile payoffs. While GPT4 displays the strongest rationality among the LLMs in the "rational environment", it does not fare as well as GPT3.5, which obtains the highest average payoffs in both the "melee" and "senior environment". Table 1 also shows for second-price auctions, Claude-I receives much higher payoffs than the rest for 20 sessions, but GPT3.5 performs the best for 100 sessions. GPT3.5 is among the LLMs that display higher rationality degree, and it also outperforms all the other LLMs in the "melee" and "senior environment".

The results indicate that while rationality degree does play a role, in order to do well in the "melee" and "senior environment", it may not be the only factor. Furthermore, while models do react to variation in opponent types, particularly when rationality assumption is not in place, changes from "melee environment" to a reduced version of it does not impact the general results much.

### 4.3 STRATEGIC REASONING THROUGH GAME HISTORY IN ECONOMICS ARENA

In a way to show strategic reasoning ability of LLMs, we reveal game history to the agents. The "rational environment" serves a baseline. Since it already emphasise to the LLMs that their opponents are rational, the presence of historical information mainly serve the purpose of learning how to improve their strategies. We also let LLMs play with one another. Through learning about other agents' past behaviours, it is expected for them to be able to reason about their strategies in view of the perceived rationality of their opponents. Altogether, we conducted 6 runs per session, and a maximum 3 runs of history were revealed to the LLMs. The history revealed is partial because of the max token constraint. However, it suffice in this context in giving some insights on convergence in actions and the strategic reasoning capability.

**Rational environment with history:** Comparing the case with history (Figure 4a) to without (Figure 1b) for beauty contests, almost all LLMs, apart from ChatGLM3 and Claude-I, show decrease in mean deviation. GPT4 has the lowest mean deviation. We can interpret that all LLMs learnt to behave closer to NE. Figure 5a shows there is convergence in action with inclusion of history. The convergence speed is substantially faster for GPT3.5 and Baichuan2 among all the models. While GPT4 behave closest to NE, its learning rate is much slower than most of the other models. This can be interpreted as GPT4, being sufficiently sophisticated, has already been rationalizing about opponents, thus providing historical information does not have as large an impact as for the rest of the LLMs. Results from ChatGLM2 and Llama2 are not recorded because they did not complete the task, implying their limited ability to take into account historical information.

In second price auctions, again comparing the case with history (Figure 4b) to without (Figure 2b), there is a general increase in mean deviation for all LLMs. This means that with historical informa-

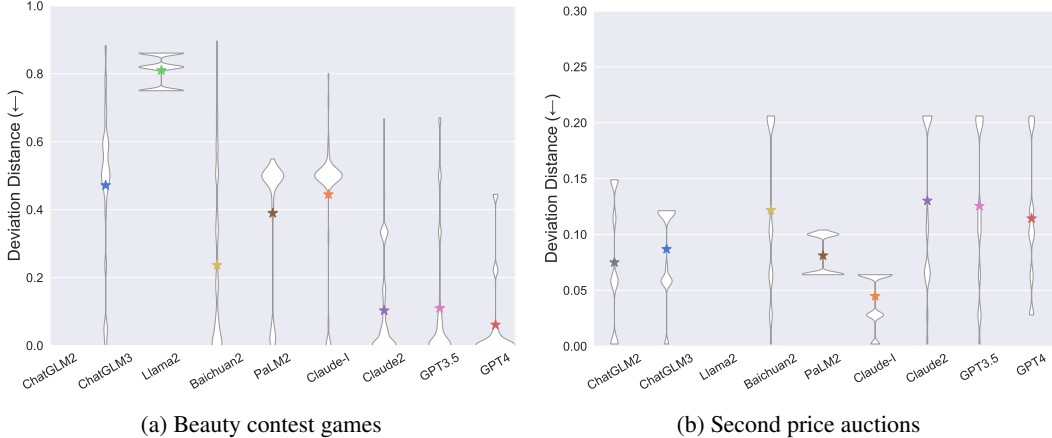

(a) Beauty contest games

(b) Second price auctions

Figure 4: Deviation distance (↓) from NEs in the "Rational environment" with history. In these experiments, we reveal a maximum 3 runs of history to the LLMs.

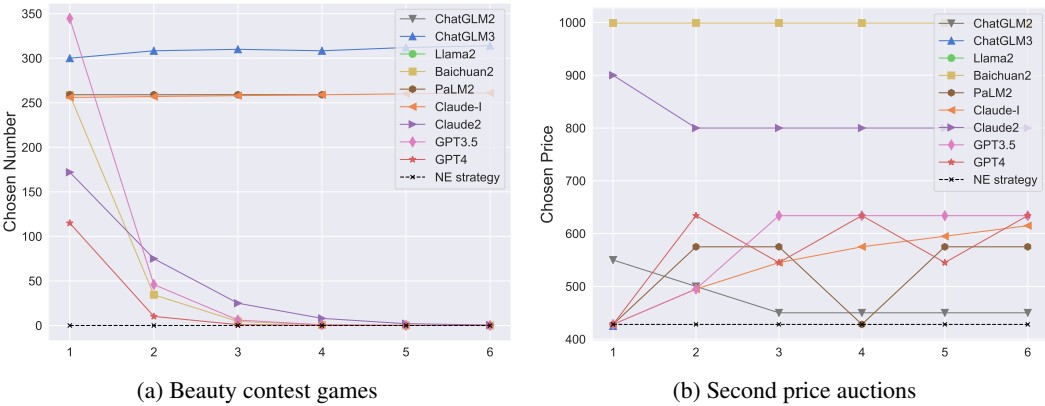

(a) Beauty contest games

(b) Second price auctions

Figure 5: The path of chosen actions over the 6 runs within a session when a maximum 3 run history is given to the LLMs, where the x-axis represents the run's index.

tion, LLMs do not adhere to their private values and tend to overbid. Among them, `Claude-I` has the lowest mean deviation, while `Claude2` has the highest. Figure 5b shows convergence in chosen action for certain models, while actions fluctuate over the runs for others. For those that shows convergence, the chosen price do not necessarily converge to private value. Once again, the absence of `Llama2` from the results suggests that it might not have the ability to consider past information.

**Melee environment with history:** When assumption of opponent rationality is not in place, which LLM fares better would be as a result of combined effect from rationality and ability to reason about others, which is facilitated through revealing history. In beauty contest games as shown in Figure 6a, `GPT3.5` receives the highest average payoffs among the LLMs, implying it has a stronger combined rationality and strategic reasoning ability than others. An interesting implication of adding history is that the top 3 performing models in (Figure 1a), ranked from `GPT3.5`, `Claude2` and `GPT4` show some decrease in average payoffs. This can be explained by the slight improvement in other LLMs. As all models are revising their strategies and improving their play, there bound to be transfer of payoffs from some models to another in the competitive game setting, the changes are indicative of LLMs attempting to reason about their opponents' strategies. The general pattern remains, models that previously achieved higher payoffs still do so.

In second price auctions, rule violations obstructed some models from producing any sensible results, thus the set of players are smaller, as in Figure 6b. Since the LLMs can reason better with history, the average payoffs are evenly spread out, with `GPT3.5` does marginally better than others.

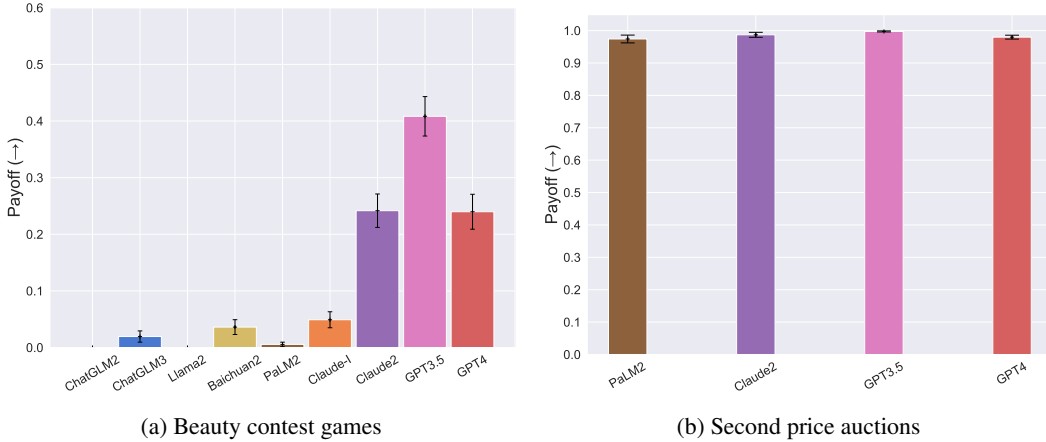

(a) Beauty contest games          (b) Second price auctions

Figure 6: Average payoffs (↑) of LLM with history.

| Game Type | History Given | LLM | | | | | | | | |
|---|---|---|---|---|---|---|---|---|---|---|
| | | ChatGLM2 | ChatGLM3 | Llama2 | Baichuan2 | PaLM2 | Claude-I | Claude2 | GPT3.5 | GPT4 |
| Beauty contest | No | 100.00 | 14.66 | 84.00 | 0.00 | 30.00 | 0.00 | 0.00 | 0.00 | 0.00 |
| | Yes | 100.00 | 6.11 | 90.00 | 0.00 | 13.89 | 12.22 | 10.56 | 3.33 | 0.00 |
| Second price auctions | No | 6.67 | 5.33 | 80.67 | 0.00 | 12.67 | 13.33 | 12.67 | 13.33 | 0.00 |
| | Yes | 4.44 | 3.33 | 78.89 | 0.00 | 11.11 | 6.67 | 7.78 | 3.33 | 0.00 |

Table 2: Frequency in the format of percentages (%) of breaking rules (↓) by different LLMs during the experiments in our economics arena. The results are obtained through tracking 150 runs without history and 180 runs with history of two kinds of game.

Through revealing game history to the LLMs, we found models to be able to leverage on past information to better their strategies. All LLMs displayed strategic reasoning capability. Models that obtain higher average payoffs in one-shot games tend to have stronger reasoning ability as well.

### 4.4 NATURAL LANGUAGE INSTRUCTION FOLLOWING BEHAVIOURS OF LLMS

During the above experiments, we also observed that certain LLMs cannot strictly follow the game instructions, i.e. fail to give responses in the specified format or break the game rules, which reflects the ability of LLMs to follow instructions in natural languages. So, we track the frequency of rule-breaking of each agent in each type of games, and the results are given in Table 2. By comparing the beauty contests and the auctions, we can see that the frequency of breaking rules is higher in the later. The rule break frequency is also higher for set-ups without history than with. Considering that prompts in auctions are more complex than the ones in beauty contests, as well as the fact that adding history makes the prompts even more complicated, we argue that certain types of LLMs fail to follow the instructions because they cannot understand the prompts with higher complexity. Therefore, we argue that our economics arena can also reflect the ability of LLMs to strictly follow the natural language instructions.

## 5 CONCLUSION

In conclusion, when LLMs were placed in competitive games without history, even when their opponents are rational, they may not behave maximally rational, and achieve lower payoffs than as dictated by the NE of the games. As we reveal the game history to the LLMs, we are testing their ability to reason about the other players' strategies, and found some displaying stronger reasoning ability that outrun the others. With history, we can also evaluate if and how fast LLMs' strategies converge to the optimal NE. When competing with themselves, some LLMs show faster convergence in action, which demonstrate faster learning ability.

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

# A   MORE DETAILS ABOUT ECONOMICS ARENA

## A.1   FORMAL DESCRIPTION OF ECONOMICS ARENA GAMES

In order to evaluate LLMs on the basis of rationality and strategic reasoning ability, we focus on multi-player games where each LLM represents one agent. By using competitive games involving multiple agents, LLMs are incentivised to select the optimal strategies to gain the best outcome by winning against other players. This strategic setting is lacking in previous games studied with LLMs (Horton, 2023; Guo, 2023; Akata et al., 2023). In this paper, we specifically focused on two types of multi-player competitive games, auctions and beauty contests, which have clearly defined NEs and comprise of actions in numeral format, making it fairly straightforward to develop statistical measures to evaluate rationality. The games are described in Figure 7. We argue that our method provides a novel approach to study not only the individual's rationality level, but also their strategic reasoning capability in rationalising their behaviour in presence of other players.

For the auction games, we use  Vickrey (1961)'s second-price sealed bid auction as the baseline. Agents were to bid for one single item, and they receive independent private signal about the value they associate with the item. They then have to submit sealed bids simultaneously. The highest bid would obtain the item and the agent has to pay the price of the second highest bid. Ultimately, the equilibrium strategy for each agent would be to bid an amount equivalent to the realisation of its private value, which is the unique symmetric Bayesian NE, and there would not be any profitable deviation[2], rendering agents irrational if that happens. Our work also attempts at varying the settings by exploring English auction where prices are called in an ascending manner and bidders are to bid sequentially. In which case, the unique symmetric equilibrium remains the same as the sealed bid auction, and the game ends when the second highest bidder drops out (Levin, 2004). By varying the setting slightly, we allow for better generalisability of the rationality results. Nonetheless, we also incorporated variations in information available to the bidders to investigate agents' strategic reasoning ability, where they best respond to other agents' bidding strategies.

The second type of games we are interested in is the beauty contest game, or otherwise named, the guessing game, we come up with a slightly modified version of the classical setting from Moulin (1986) and Nagel (1995). Herein, agents are asked to choose a number between the interval 0 and $\bar{c}$, where $\bar{c} = 100$ in the classic model, but we allow this to vary to prevent agents from learning through reinforcement of past experiences. They were informed that the one who selects a number closest to $\frac{2}{3}$ of the average of all chosen numbers will win the game, and obtain a fixed prize of $\$x$ (i.e. $x \in \mathbb{R}^+$), while the others receive 0 payoff. In case of a tie, the prize will be split amongst those who tie. In this game, one not only assumes all players are rational, a stricter assumption of common knowledge of rationality is in place, which involves iterative elimination of dominated strategies, leading to a unique NE where all agents play 0 (Yildiz, 2012). Playing strictly dominated strategies are considered irrational, thus the degree of deviation away from NE can be interpreted as the level of irrationality. Furthermore, in experiments with human subjects, Nagel (1995) found out-of-the equilibrium behaviour and proposed the possibility of agents having finite depth of reasoning ability. In relation to testing on LLMs, we can first determine the level of reasoning for each LLM in the static game, and by revealing information about past games, this would allow us to test for agents' strategic reasoning, as well as learning ability, in finding the best response given their perception about the other players' guesses.

## A.2   ARCHITECTURE OF ECONOMICS ARENA

Experiments in `EconArena` can be easily run through: 1) configuring a `TOML` file to specify all hyperparameters for the games; 2) setting up a session consists of a certain number, e.g 50, of independent runs of the game; 3) running the game by initialising and interacting with players, of which each is backboned by a newly instantiated LLM model. The results reported by `EconArena` will contain both the logs of independent runs, and the log for the whole session. More details about the architecture of our implementations are shown as follow:

---

[2]Except in cases of asymmetric equilibrium, which are not investigated to avoid diving into the difficulty of coordination and equilibrium selection.

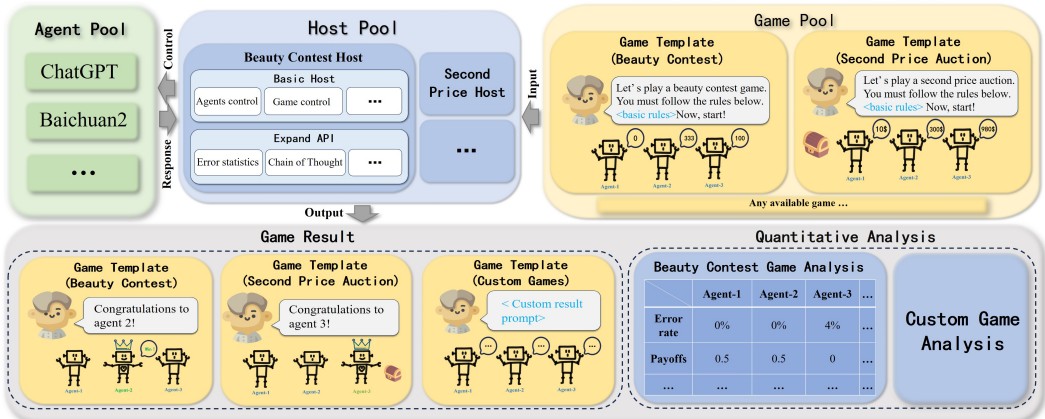

Figure 7: Diagram of `EconArena` which is constituted by three major modules: i) hosts; ii) agents; and iii) games. The `hosts` are responsible for running the games, interacting with the LLMs through APIs, collecting and returning game results. The `agents` are wrappers of APIs of various LLMs, and the `games` describe the rules of various economics games.

**System Design:** There are the following components in the system: 1) Host, which is the main class that hosts all the economic games and the agents; 2) Game, which provides the game environments and the rules of the games. Note that only host can interact with the games; 3) Agent, which encapsulates the LLMs' interfaces and provides the methods for the host to interact with the various backbone LLMs. In the following subsections, we'll describe the design of each component in detail.

**Hosts:** 'hosts/hosts.py' implements the class 'Host' for the initialization and running of the auction games. It has the ability to interact with both the 'Agents' side and 'Games' side, including functions: 1) initial the game; 2) receive the initial prompts generated by 'Games' and forward them to the 'Agents'; 3) receive the raw responses from the 'Agents' and deduce them to simple float numbers and forward the results to 'Games'; 4) receive terminal flag and rule-breaking flags from 'Games' to update the game running state; 5) send specific prompts every round to the 'Agents'; 6) receive final results generated by 'Games'.

To implement all these functions, following interfaces are developed: 1) '__init__': initialize the host; 2) '_update_prompt': update the specific prompt after receive the status of the last round; 3) 'game_running': run the multi-round (including one single round situation) action game.

**Games:** Variants economics games are implemented in the 'game' module, each kind of game is implemented as many classes in a separate file, e.g. the auction games in 'game/auctions.py'. Each kind of games are described in the following subsections. The common interfaces of the games are the following: 1) '__init__': initialise the game environment, which should be called by the Host classes before the game starts; 2) 'get_init_prompts': the method for the host to get the initial prompts of the auction games. It returns a list of the prompts for the players to understand the rules of the auction. Note that the prompts are in the same order as the players. 3) 'check_bidding': the method for the host to check the bidding prices of players. It takes a list of the bidding prices of players and returns a boolean value indicating whether the bidding prices are valid. Note that the bidding prices are in the same order as the players. 4) 'bid': the method for the host to pass the bidding prices of players to the auction games. It returns a boolean value indicating whether the game is finished after the current round. 5) 'get_results': the method for the host to get the results of the games. The results vary across different kinds of games, which will be described in the following subsections.

- **Auction Games:** The auction games are implemented in 'game/auctions.py', and the 'get_results' interface is explained in the following: 1) 'get_results': returns a list of the results of the auction games including whether the auction is valid, the total number of rounds, the assets and the payoffs of the players after the auction games are finished, as well as the payoffs of the players under Nash equilibrium. Note that the assets and the payoffs are in the same order as the players, but have different meanings. 2) 'assets': the fraction of the assets that the players still have after the auction

games are finished over the initial assets. The asset is the total value of the cash a player holds and the private value of the bidding items to it. Note that the assets will be deducted an entrance fee if a player breaks the rules of the auction games. Suppose a player's initial asset is 100, and it gets the item whose private value is 60 to it with a bidding price of 50, then the asset of the player after the game is 110. 3) 'payoffs': the fraction of the assets that a player still have after the auction games are finished over the theoretical Nash equilibrium asset of that player. Suppose the player should have an asset of 100 at the Nash equilibrium, but it gets an asset of 110 after the game, then the payoff of the player is 1.1.

- · **Contest Games:** The contest games are implemented in 'game/contests.py', and the 'get_results' interface is explained in the following: 1) 'get_results': returns a list of the winners of the contest games, a list of the payoffs of all players after the games are finishes, and a list of the payoffs of all players under the Nash equilibrium. The payoffs and NE payoffs are in the same order as the players.

**Agents:** 'agent/agents.py' implements an abstract class 'LLMAgent' for all the agents based on LLMs, it also implements all subclasses of 'LLMAgent' for the specific LLMs such as 'GPT4' and 'PaLM2'.

We first describe the abstract class 'LLMAgent' and the abstract methods in it. 1) '__init__': this method is to initialise the objects. Since the backbones are all LLMs, we need to specify the API key. We may also need tokenisers for preprocessing the input. 2)'act': this method is to interact with the API of the LLMs. It first preprocesses the input prompt with the following 'preprocess' function, then calls the API of the LLMs and get the response via 'get_response', and finally postprocesses the output with the following 'postprocess' function before returning it. 3) 'preprocess': this method is to preprocess the input prompt. It takes the input prompt and returns the processed prompt. Various LLMs may have different preprocessing methods, which is part of the prompt engineering we have to do. 4) 'get_response': this method is to interact with the LLMs via either APIs or local service. It takes the processed prompt and returns the response from the API of the LLMs or through the local interface with the LLMs. 5) 'postprocess': this method is to postprocess the output response. It takes the output response from the API of LLMs and returns the processed response. Various LLMs may have different postprocessing methods, which is again part of the prompt engineering we have to do. 6) [Optional] 'add_to_memory': this is method is to manage the memory pool of LLM agents. For example, in some cases, we can store the game history to improve the performance of the LLMs. It is optional, since memoryless agents can also work well in some cases.

### A.3 DISCUSSION IN FURTHER DETAILS ABOUT VARIOUS METRICS IN ECONOMICS ARENA

### A.3.1 RATIONALITY DEGREE

Rationality concept has been used rigorously in economics and forms the basis of many neoclassical economic models (Sentiments, 2019). Since LLMs are pre-trained on human produced data, it is anticipated to be subjected to some extent of human biases. However, if and how much it differs from human-like behaviour would need an explicit measure, for which economics games provide a good evaluation environment.

Given that the rationality assumption holds, the optimal strategy would be the NE. As per equilibrium definition, any deviation away from the NE is less profitable. In the game simulation with LLM agents, if all of them are rational, their payoffs should converge to the NE payoffs, or we say the optimal payoffs. Following which, we define a measure

$$r_i = \frac{\frac{1}{T} \sum_{t=1}^{T} \rho(a_{it})}{\rho_{op}} \tag{1}$$

where $a_{it}$ is the action (a bidding price or a number) chosen by agent $i$ at time stamp $t$; $\rho(a_{it})$ equals to payoff after the action $a_{it}$; $\rho_{op}$ refers to optimal payoff of the respective game; $T$ indicates the total time; $r_i$ refers to the ratio of average payoff of agent $i$ over total time $T$ to the optimal payoff.

However, in practice, it is possible that not all LLMs are completely rational players. Thus, we further propose self-competing games, in which agents compete with others backed by the same

type of LLM, and the deviation distance of LLM's strategies away from NE is used as a measure for their respective rationality level. In self-competing game, we define deviation from NE as

$$\overline{d} = \frac{1}{nT} \sum_{i=1}^{n} \sum_{t=1}^{T} d_{it} \tag{2}$$

$$d_{it} = \left| \frac{\pi(a_{it})}{\pi(a_{it}^*)} - 1 \right| \tag{3}$$

where $a_{it}$ is the action chosen (a bidding price or a number) by agent $i$ at time $t$; $a_{it}^*$ is the NE action of agent $i$ at time $t$; $\pi$ is a payoff function related to game: in second-price auction game $\pi(a_{it})$ equals to asset after the action $a_{it}$, while in beauty contest game $\pi(a_{it})$ equals to the value associated with the action $a_{it}$ itself; $d_{it}$ is the deviation from NE of agent $i$ at time $t$; $\overline{d}$ is the average deviation from NE across time periods. Smaller $\overline{d}$ implies better rationality in specific game, as well as lower frequency of irrational behaviours. Moreover, when $\overline{d}$ of some LLMs are similar to each other, the more centralised the distribution of $d_{it}$, the more consistent the rationality performance of the LLM.

In both the auction games and the beauty contest games, the unique NE is clearly defined, we can therefore quantify the deviation in each case, providing a fairly straightforward metric to evaluate one's rationality. It is particularly noted that the use of such measurement was not exposed to the LLMs playing the games. Furthermore, the special characteristic of competitive games dictates that while agents aspire to maximise one's payoff, they also hope to win, these could be conflicting objectives. Thus, it is necessary for us to specify whether the LLMs need to be strictly rational in prompts, otherwise, it might be difficult to filter out if the action was chosen because one wants to win at all cost, or one is simply being irrational.

### A.3.2 STRATEGIC REASONING ABILITY

In competitive games, we can expect agents to be playing the NE strategies when there is common knowledge of rationality. However, when it is possible that opponents are not completely rational, the rational strategy might not necessarily be the winning strategy. As a result, in order to gain the highest payoff, a player would need to reason about the strategies of other players, which we defined to be their strategic reasoning ability. Given historical information, we expect that models who are capable of strategic reasoning to show timely adaption to the strategies of other players to avoid loss or even increase payoff. The higher payoffs one is able to obtain over time, the better one is able to form correct beliefs about other players' strategies, and thus the greater is one's strategic reasoning ability. We define a similar metric as the measure in 1

$$r_i = \frac{\frac{1}{T} \sum_{t=1}^{T} \rho(\hat{a}_{it})}{\rho_{op}} \tag{4}$$

where $\hat{a}_{it}$ is the action chosen by agent $i$ at time stamp $t$ (with history given); $\rho(\hat{a}_{it})$ equals to payoff after the action $\hat{a}_{it}$; $\rho_{op}$ refers to optimal payoff of the respective game; $T$ indicates the total time; $r_i$ refers to the ratio of average payoff of agent $i$ over total time $T$ to the optimal payoff. The closer the distance between the ratio and 1, the better the strategic reasoning ability.

In order to further distinguish the level of LLMs' strategic reasoning ability, we vary the completeness of historical information[3] mainly in two ways:

1) Only the strategies of other players in the past games (applies to both game types)

2) Provides the private information along with the strategies of other players in the past games (applies to the auction games)

Our assumption is that the more information is revealed, the agent with stronger strategic reasoning ability is more likely to win. Since the beauty contest game has a more simplified setting, it is

---

[3]The history is not necessarily in the same session of the games, details will be specified in the experimental setting

anticipated that by showing the strategies of other players, it would be sufficient to distinguish the stronger player. Whereas for the auction games, they are more complex, thus more information might be necessary for better identification, on the other hand, insufficient information may mislead LLMs to pursue wrong strategies.

Moreover, we take irrational behaviours (i.e. overbids) as well as bad strategies (i.e. miscalculations) into account and suggest that a model with good strategic reasoning ability may not obtain high expected payoffs due to such factors. As a result, we also make use of self-competing games, and specifically inform LLMs that all the players are of similar level of rationality and strategic reasoning ability. By force of contrast, we could potentially deduce which models are the ones that should have received higher payoffs when there are less unfavourable factors. Meanwhile, in self-competing beauty contest game, we can observe whether the curve of respective LLM's strategies converges to the NE in order to further verify their strategic reasoning ability.

## B    MORE DETAILS ABOUT EXPERIMENT SETUPS

### B.1    BEAUTY CONTEST IN THE MELLE ENVIRONMENT

In this setup, different types of LLMs play against each other in one-run beauty contest games.

Below is the prompt of the experiment for LLM agents.

---

**[SYSTEM]**
I want you to act as a clever game player and perform as rationally as possible.

**[USER]**
You are playing a game and the rules are as follows:

- it consists of *{number of players}* players, including you.
- this is a one-round game.
- you need to choose a real number between 0 and *{upper bound}*, 0 and *{upper bound}* inclusive.
- everyone does not know how others would make choices beforehand.
- the winner is the person whose number is the closest to the two thirds of the average of all chosen numbers.
- the winner gets a fixed prize, in case of a tie the prize is split amongst those who tie.
- your goal is primarily to maximize the possibility of getting the prize and secondly to maximize the your prize.

Please just strictly output a JSON string, which has following keys:

- understanding: str, your brief understanding of the game
- popular answer: float, the number which you think other players are most likely to choose
- answer: float, the number which you would like to choose
- reason: str, the brief reason why you give the popular answer and the answer that way

---

### B.2    BEAUTY CONTEST IN THE RATIONAL ENVIRONMENT

In this setup, one certain type of LLM plays against others who are all hard-coded rational agents in one-run beauty contest games. **Note that the LLM agent is informed of opponent rationality.**

Below is the prompt of the experiment for LLM agent.

---

**[SYSTEM]**
I want you to act as a clever game player and perform as rationally as possible.

**[USER]**
You are playing a game and the rules are as follows:

- ...(same as the melle environment)
- you can assume that other are all perfectly rational players.

Please just strictly output a JSON string, which has following keys:

- ...(same as the melle environment)

---

## B.3 BEAUTY CONTEST WITH CHAIN OF THOUGHT TECHNIQUE

In this setup, one certain type of LLM plays against others who are all hard-coded rational agents in one-run beauty contest games. **Note that the LLM agent is informed of opponent rationality and is asked to use chain-of-thought to think step by step.**

Below is the prompt of the experiment for LLM agent.

---

**[SYSTEM]**
I want you to act as a clever game player and perform as rationally as possible.

**[USER]**
You are playing a game and the rules are as follows:

- ...(same as the melle environment)
- you can assume that other are all perfectly rational palyers.

Let's think step by step.

After that, please output a JSON string, which has following keys:

- popular answer: float, the number which you think other players are most likely to choose
- answer: float, the number which you would like to choose

---

## B.4 BEAUTY CONTEST WITH HISTORICAL INFORMATION

In this setup, different types of LLMs play against each other in **multi-runs** beauty contest games **with** history information given.

Below is the prompt of the experiment for LLM agent.

---

**[SYSTEM]**
I want you to act as a clever game player and perform as rationally as possible.

**[USER (run 1)]**
You are playing a game and the rules are as follows:

- ...(same as the melle environment)

Please just strictly output a JSON string, which has following keys:

- ...(same as the melle environment)

**[USER (runs after run 1)]**
The game of the same config has been hold for {*number of runs*} run(s), and the historical choices of everyone are shown below (your id is {*ID of the player*}):
{*historical information (in JSON format)*}
Everyone can optimize his/her answer with the history to play in a new run in order to achieve goals.
Please just strictly output a JSON string for a new run, which has following keys:

- goal: str, briefly check if you still remember what goal you should achieve in the game
- previous answer: float, the number which you chose in the last run
- answer: float, the number which you would like to adjust your choice to
- reason: str, the brief reason why you adjust the answer that way

---

## B.5 SECOND-PRICE AUCTION IN THE MELLE ENVIRONMENT

In this setup, different types of LLMs play against each other in one-run second-price auctions.

Below is the prompt of the experiment for LLM agents.

---

**[SYSTEM]**
I want you to act as a smart auction bidder and perform as rationally as possible.

**[USER]**
You are participating in an auction and the rules are as follows:

- it consists of *{number of bidders}* bidders, included you.
- this is a one-round auction.
- there is only 1 item, and your private value of the item is *{private value of the bidder}* units(private values may vary among bidders).
- you have *{assets of the bidder}* units of assets, and you need to place a bid which is not higher than your assets.
- everyone does not know either private value of others or how others would make choices beforehand.
- the bidder who places the highest bid among all the bids will get the item(others will not), and only need to pay an amount of assets equalling to the second-highest bid among all the bids.
- if there are multiple highest bid, only the bidder with the minimal id will get the item.
- if you get the item, your payoff equals to your remaining assets(=assets deducting payment) plus your private value, otherwise your payoff equals to your original assets.
- your goal is to maximize your overall payoffs(notice that getting the item is not necessary) considering the situation unpredictable.

Please just strictly output a JSON string, which has following keys:

- understanding: str, your brief understanding of the auction
- bid: float, the bid which you would like to place
- reason: str, the brief reason why you place the bid that way

---

## B.6 SECOND-PRICE AUCTION IN THE RATIONAL ENVIRONMENT

In this setup, one certain type of LLM plays against others who are all hard-coded rational agents in one-run second-price auctions. **Note that the LLM agent is informed of opponent rationality.**

Below is the prompt of the experiment for LLM agent.

---

**[SYSTEM]**
I want you to act as a smart auction bidder and perform as rationally as possible.

**[USER]**
You are participating in an auction and the rules are as follows:

- ...(same as the melle environment)
- your goal is to maximize your overall payoffs(notice that getting the item is not necessary)
- you can assume that others are all perfectly rational bidders.

Please just strictly output a JSON string, which has following keys:

- ...(same as the melle environment)

---

## B.7 SECOND-PRICE AUCTION WITH CHAIN OF THOUGHT TECHNIQUE

In this setup, one certain type of LLM plays against others who are all hard-coded rational agents in one-run second-price auctions. **Note that the LLM agent is informed of opponent rationality and is asked to use chain-of-thought to think step by step.**

Below is the prompt of the experiment for LLM agent.

> **[SYSTEM]**
> I want you to act as a smart auction bidder and perform as rationally as possible.
>
> **[USER]**
> You are participating in an auction and the rules are as follows:
>
> - ...(same as the melle environment)
> - your goal is to maximize your overall payoffs(notice that getting the item is not necessary)
> - you can assume that others are all perfectly rational bidders.
>
> Let's think step by step.
> After that, please output a JSON string, which has following keys:
>
> - bid: float, the bid which you would like to place

## B.8 SECOND-PRICE AUCTION WITH HISTORICAL INFORMATION

In this setup, different types of LLMs play against each other in **multi-runs** second-price auctions **with** history information given.

Below is the prompt of the experiment for LLM agent.

> **[SYSTEM]**
> I want you to act as a smart auction bidder and perform as rationally as possible.
>
> **[USER (run 1)]**
> You are participating in an auction and the rules are as follows:
>
> - ...(same as the melle environment)
>
> Please output a JSON string, which has following keys:
>
> - ...(same as the melle environment)
>
> **[USER (runs after run 1)]**
> The auction of the same configuration has been hold for {*number of runs*} run(s), and the historical information of everyone are shown below (your id is {*ID of the bidder*}):
> {*historical information (in JSON format)*}
> Everyone can optimize his/her bid with the history to place bid in a new run in order to achieve the goal.
> Notice that everyone's original asset and private value will stay unchanged.
> Please just strictly output a JSON string for a new run, which has following keys:
>
> - goal: str, briefly check if you still remember what goal you should achieve in the game
> - previous bid: float, the bid which you placed in the last run
> - previous payoff: float, the payoff which you got in the last run
> - bid: float, the bid which you would like to adjust to
> - reason: str, the brief reason why you adjust the bid that way

## C  THE LIMITATIONS OF THIS WORK

While we explored the two types of competitive games and show interesting results, we also recognise that the number of games included in this paper is limited, and the variations in the set-ups are restricted. There are potential to generalise and refine our metrics further by investigating more forms of games.

Another possible limitation lays in the sensitivity of results to key elements in prompts. For instance, LLMs' behaviours could be strongly reliant on the explicit requirement of rationality in the prompts, without which, LLMs might have other objectives, such as winning the games at all cost, that could lead to conflicting results or higher frequency of rule breaking. Furthermore, it is also possible that our observations are restricted to the language environment that was picked. Following studies on Wittgenstein's formalisation of language as a set of rules or practices (Malcolm, 1989), human following different rules could behave differently as according to their language settings. Since LLMs are trained on large amount of human-generated data, by varying the language prompts, we could effectively separate and compare the metrics measure by different groups of "rule-followers". The degree of rationality and strategic reasoning ability could differ for the same LLM across the language settings, which could provide another dimension to evaluate the LLMs. While this has not been analysed in this paper, it is an interesting direction and is intended to be explored.

Last but not the least, a question that is still open to discussion is how we can combine the game idea with the real world applications. In practical scenarios, we often do not have the ground-truth, (i.e. the knowledge of NE strategies), therefore it would be difficult to analyse how rational LLMs are in those situations. However, the economics games provide a simplified environment to evaluate the models, and could be compared with the experimental results of human subjects to check if they could perform better, more rationally and more strategically. This might highlight a case for real life application, where following advice of certain more advanced LLMs could lead to better outcome. Nonetheless, these games serve as the basis before evaluating more complicated models that have theoretical predictions and closer to real world scenarios.

