# OpenReview forum: "Large Language Models as Rational Players in Competitive Economics Games"
_ICLR.cc/2024/Conference — Submitted to ICLR 2024_

### Official Review · Reviewer_xNRn · 2023-10-24

**Soundness:** 1 poor
**Presentation:** 2 fair
**Contribution:** 1 poor
**Rating:** 3
**Confidence:** 5

**Summary:**

In this paper, the authors propose to evaluate LLM using two standard games: second-price auctions and beauty games. The author measure the performances using the distance to their canonical Nash equilibria. The author also provide experimental results.

**Strengths:**

The idea of using games to using LLMs is promising.

**Weaknesses:**

There are certain flaws of the approach. Particularly on the point of using stage Nash equilibrium to evaluate LLMs in a repeated game setting.

**Questions:**

While I like the idea of using game to evaluate LLMs, I think there are certain flaws of this paper's approach. Particularly, the author claim the distances to Nash equilibria in these games are reasonable measures. While for a one-shot version of these game it does make sense to some extent, I believe this is no longer true for repeated version of these games. The folk theorem in repeated games indicates the set of Nash may far more larger than stage-Nash. For example in the iterated Prisoners' dillemma, the stage-Nash is (defect, defect), but people may usually regard tit-for-tat as a rational strategy. So for repeated games, the problem of equilibrium selection becomes harder.


Another question is about the choices of games. While these two games are very classical, this definitely limited the settings. Furthermore, isn't it possible that the LLMs may already knows like the optimal strategies in these games, given that they are very famous?

---

> ### Author Response · Authors · 2023-11-21
>
> Thanks for recognising the potential of this work. Regarding the reviewer's concerns, we reply as follows.
>
> 1. Measure based on NEs
>
> > While I like the idea of using game to evaluate LLMs, I think there are certain flaws of this paper's approach. Particularly, the author claim the distances to Nash equilibria in these games are reasonable measures. While for a one-shot version of these game it does make sense to some extent, I believe this is no longer true for repeated version of these games. The folk theorem in repeated games indicates the set of Nash may far more larger than stage-Nash. For example in the iterated Prisoners' dillemma, the stage-Nash is (defect, defect), but people may usually regard tit-for-tat as a rational strategy. So for repeated games, the problem of equilibrium selection becomes harder.
>
> Thanks for highlighting this issue. For most part of the paper, we are doing multiple sessions of one-shot games, since an agent’s current period action has no implication on future actions, we can use deviation away from NE as one of the measures. In the case of revelation of historical information, we understand that there could be some influence across different runs, though not in the traditional sense as in repeated games. The main focus here would be the convergence of strategies to demonstrate that agents are able to reason about other players’ strategies and learn from past information. While we loosely define rational strategy in this case to be the same as NE strategy in a stage game, the best strategy that generates the highest payoffs may not be that strategy, therefore, the combination of metrics highlighted in Section 3.2 are needed to evaluate the LLMs.
>
> 2. Choice of games
>
> > Another question is about the choices of games. While these two games are very classical, this definitely limited the settings. Furthermore, isn't it possible that the LLMs may already knows like the optimal strategies in these games, given that they are very famous?
>
> Thanks for proposing the question. As shown in our updated results, if LLMs already know the optimal strategies, they shouldn’t deviate from NEs when the prompt clearly states that the opponents are rational and clearly asks them to behave in the most rational manner.
>
> A very nice feature of our EconArena is that the environment can be dynamicise in several different dimensions, e.g. the configurations of the games, the setup of the players, and the overall rationality degree of the players. Compared with static benchmarks, such dynamic setups are very unlikely to be seen during the training of LLMs, especially considering that new LLMs are rolled out these days.

---

### Official Review · Reviewer_xpxk · 2023-10-28

**Soundness:** 2 fair
**Presentation:** 3 good
**Contribution:** 1 poor
**Rating:** 3
**Confidence:** 4

**Summary:**

This paper aims to assess the strategic reasoning and rationality of large language models (LLMs) in competitive games, specifically focusing on the second price auction and beauty contest game. The paper compares the performance of various versions of GPT-4 and GPT-3.5.

**Strengths:**

1. The paper is well-written and organized, making it easy to follow.
2. Investigating the capability of strategic reasoning in LLMs is important.
3. The choice of using the second price auction and beauty contest game as an evaluation arena for LLMs is novel.

**Weaknesses:**

1. The significance of this paper appears to fall short of the standard of ICLR. As acknowledged by the authors, there are already a few related works that explore the performance of LLMs in economic games with different settings, including various game classes and information provided to LLMs. It is already known that LLMs can exhibit some degree of rationality and strategic reasoning. Although this paper evaluates LLMs under specific new settings, it does not significantly advance our understanding of LLMs in economic games.

2. As a paper that selects "dataset and benchmark" as the primary area, the experiments are not thorough enough to support the main claims and reproducibility is questionable. Most importantly, the prompts which are central for reproducibility are not presented.  Moreover, only multiple versions of GPT are considered, neglecting other commonly used LLMs (e.g., Claude 2 or LLama 2) which have been shown to have quite different performances in games compared to GPT, and only the results of average values are reported (e.g., Figure 2 and Figure 4).  Additionally, the paper claims that they have varied the prompts to show the robustness of their results, but these experiments are not presented.

**Questions:**

What is the reason for choosing the second-price auction and beauty contest game rather than other zero-sum games, such as matching pennies and rock paper scissors?

---

> ### Author Response · Authors · 2023-11-21
>
> We thank the reviewer for identifying the novelty of this work. We reply to the reviewer's concerns one-by-one below.
>
> 1. Significance concern
>
>  > The significance of this paper appears to fall short of the standard of ICLR. As acknowledged by the authors, there are already a few related works that explore the performance of LLMs in economic games with different settings, including various game classes and information provided to LLMs. It is already known that LLMs can exhibit some degree of rationality and strategic reasoning. Although this paper evaluates LLMs under specific new settings, it does not significantly advance our understanding of LLMs in economic games.
>
> We have updated the methodology and experiment sections, and they have fixed the reviewer’s concerns about the significance of this work.
>
> 2. Dataset and benchmark
>
> > As a paper that selects "dataset and benchmark" as the primary area, the experiments are not thorough enough to support the main claims and reproducibility is questionable. Most importantly, the prompts which are central for reproducibility are not presented. Moreover, only multiple versions of GPT are considered, neglecting other commonly used LLMs (e.g., Claude 2 or LLama 2) which have been shown to have quite different performances in games compared to GPT, and only the results of average values are reported (e.g., Figure 2 and Figure 4). Additionally, the paper claims that they have varied the prompts to show the robustness of their results, but these experiments are not presented.
>
> We believe the updated version fixed all these concerns from the reviewer.

---

### Official Review · Reviewer_8dxE · 2023-10-30

**Soundness:** 2 fair
**Presentation:** 3 good
**Contribution:** 1 poor
**Rating:** 3
**Confidence:** 3

**Summary:**

This paper proposed to explore competitive games as an evaluation of the rationality and strategic reasoning ability of LLMs. By varying the game history revealed to LLMs-based players, they found that most LLMs are rational in the sense of playing strategies that can increase their payoffs, but not the most rational strategies, i.e., Nash Equilibria (NEs). Moreover, when game history is available, certain types of LLMs can converge faster to the NE strategies. Other abilities of LLMs were tested.

**Strengths:**

This paper proposed to explore competitive games as an evaluation of the rationality and strategic reasoning ability of LLMs. They can be used to test the abilities of LLMs, i.e., rationality, strategic reasoning ability, and instruction-following capability.

**Weaknesses:**

Competitive games exist in the literature. This paper just shows how to design experiments to test the ability of LLMs. That is, this paper did not provide any dataset or benchmark, and then it should not be added to the primary area of datasets and benchmarks as well.

This paper is not a technical paper on learning representation. Thus, it is not related to ICLR.

**Questions:**

.

---

> ### Author Response · Authors · 2023-11-21
>
> > Competitive games exist in the literature. This paper just shows how to design experiments to test the ability of LLMs. That is, this paper did not provide any dataset or benchmark, and then it should not be added to the primary area of datasets and benchmarks as well.
> >
> > This paper is not a technical paper on learning representation. Thus, it is not related to ICLR.
>
> We want to kindly remind the reviewer that “applications in economics” is part of the subject areas of ICLR, and evaluation methods are definitely of the interest of ICLR audience.
>
> Regarding the existence of competitive games, we know there is concurrent work on auction arena, but this is no pre-existing literature on evaluating LLMs with competitive economics games. Can the reviewer provide a reference to back up the claim “competitive games exist in the literature”?
>
> Moreover, as we stated in our updated paper, we will release the EconArena as both a Python package publicly available on PyPI and a website where users can config and run games with GUI operations. This kind of dynamic “benchmark” is definitely a kind of dataset/benchmark.

---

### Official Review · Reviewer_HCDQ · 2023-11-03

**Soundness:** 3 good
**Presentation:** 2 fair
**Contribution:** 2 fair
**Rating:** 3
**Confidence:** 4

**Summary:**

The paper investigates the strategic reasoning abilities of Large Language Models (LLMs) by employing them as rational agents in competitive economic games. The study focuses on the second-price auction and beauty-contest games. It also includes a variation, the self-computing beauty contest game, to analyze the LLMs' ability to adapt strategies based on other agents' behaviors when those agents are instances of the same model. GPT-4, among other models, is highlighted for its capacity to quickly converge to the Nash Equilibrium, demonstrating a strong capability for strategic adjustment and reasoning in this specific case.

**Strengths:**

* Arguing about metrics that associate with the Nash Gap is a systematic approach. This is indeed a straightforward metric to evaluate rationality.
* The use of self-computing beauty contest games to assess LLMs is an interesting one, compared to having different models in the same auction.
* The methodology includes running multiple runs and account for the LLMs not responding correctly, an inherent pitfall of their architecture.

**Weaknesses:**

* The paper acknowledges the importance of prompt sensitivity but fails to provide a detailed account of prompt structures, limiting the reproducibility of the experiments. It does not investigate different methods of prompting or incorporating historical data into prompts.
* There is no discussion on whether the stability of outputs in the homogeneous model setting correlates with a consistent strategy distribution between all agents, nor is there an exploration of how the model's temperature setting influences strategy uniformity.
* The methodology is unclear on whether different rounds were obtained by a chat-based model instantiation, where there are affecting subsequent decisions, or if the models were independently assessed in varied historical contexts, simulating different rounds.
* The experiments lack scenarios where LLMs interact with actual strategic agents, which would test the models in more realistic strategic environments.

Miscellaneous:
* The font size in Figure 1 is difficult to read.

**Questions:**

* How does the variation in prompt structure affect the strategic decision-making of LLMs, and what would be a good prompt structure to accurately assess their performance?
* Is the sample size of rounds (e.g., 10 rounds) statistically significant enough to establish the rate at which GPT-4 converges to the optimal strategy? Could one argue that gpt-4 is learning with some arbitrary learning rate over time?
* What role did the temperature parameter play in the stability of output in games where all models were the same? The paper argues about gpt-4 facing the inconsistency other models. For a given temperature parameter there would also be self-inconsistency. Also, as LLMs are rolled out into ever more realistic scenarios, the assumption of competing with the same agent is not realistic.
* How were the answers considered? Was it the next token in the generation process, or a number found after allowing the LLM to complete the sentence until reaching the <eos> token?
* What are the effective ways to prompt LLMs to account for historical information, and how can this be standardized to assess performance sensitivity to prompt phrasing? Since there is sensitivity in the prompt and as the history becomes larger, how is the information of the history compressed into the prompt?
* Can the strategic behaviors observed in LLMs within the confines of the study be generalized to other forms of competitive games or economic models?

---

> ### Author Response · Authors · 2023-11-21
> **Reply to Reviewer HCDQ (1/2)**
>
> We thank the reviewer for pointing out the interesting question this work tries to answer. We reply the reviewer’s concerns one-by-one below.
>
> 1. Details about prompts
>
> > The paper acknowledges the importance of prompt sensitivity but fails to provide a detailed account of prompt structures, limiting the reproducibility of the experiments. It does not investigate different methods of prompting or incorporating historical data into prompts.
>
> Thanks for brining this up, we provided all the prompts we used in the updated Appendix B.
>
> 2. Stability and uniformity of LLMs’ strategies
>
> > There is no discussion on whether the stability of outputs in the homogeneous model setting correlates with a consistent strategy distribution between all agents, nor is there an exploration of how the model's temperature setting influences strategy uniformity.
> What role did the temperature parameter play in the stability of output in games where all models were the same? The paper argues about gpt-4 facing the inconsistency other models. For a given temperature parameter there would also be self-inconsistency. Also, as LLMs are rolled out into ever more realistic scenarios, the assumption of competing with the same agent is not realistic.
>
> Thanks for pointing this out. In our updated version, we use the self-compete set up for sanity check to verify consistency in strategies.
>
> 3. Instantiation of the LLM backbones
>
> > The methodology is unclear on whether different rounds were obtained by a chat-based model instantiation, where there are affecting subsequent decisions, or if the models were independently assessed in varied historical contexts, simulating different rounds.
>
> Thanks for bringing this up. We’ve fixed this problem in the updated Section 4. In short, yes, for each independent run and thus session of the same game, we re-instantiate a new interface with the LLM.
>
> 4. Strategic agents
>
> > The experiments lack scenarios where LLMs interact with actual strategic agents, which would test the models in more realistic strategic environments.
>
> Thanks for the suggestions. In our updated version, we established several agents with hard-coded rational or irrational strategies, and showed the results of LLMs competing with these different baselines. Other pre-defined strategies were not included at the moment. Since the paper mainly focused on exploring competition between different LLM-based agents, actual strategic agents were not considered currently, but could serve as interesting extension.
>
> 5. Effect of prompt engineering on the performance of LLMs
>
> > How does the variation in prompt structure affect the strategic decision-making of LLMs, and what would be a good prompt structure to accurately assess their performance?
>
> Thanks for this good question! We agree that the prompt structure may affect the strategic decision-making of LLM, thus we decided to introduce the least amount of prompt engineering into this work. In the meantime, we have also set up the APIs for future users of EconArena to customise their own prompts for all games. A detailed discussion about the optimal prompt structure is not the focus of this work, as our aim is to provide the community a public economics environment for evaluating LLMs.
>
> 6. Effect of the Number of rounds on the convergence rate
>
> > Is the sample size of rounds (e.g., 10 rounds) statistically significant enough to establish the rate at which GPT-4 converges to the optimal strategy? Could one argue that gpt-4 is learning with some arbitrary learning rate over time?
>
> Thanks for pointing out this ambiguity. We further clarify the convergence experiment setup in the updated Section 4.
>
> 7. Answer post-processing
>
> > How were the answers considered? Was it the next token in the generation process, or a number found after allowing the LLM to complete the sentence until reaching the <eos> token?
>
> Thanks for the question. We didn't post-process the answers returned from LLMs, as we specified the format that LLMs should follow in the prompts.
>
> 8. Prompt engineering for game history
>
> > What are the effective ways to prompt LLMs to account for historical information, and how can this be standardized to assess performance sensitivity to prompt phrasing? Since there is sensitivity in the prompt and as the history becomes larger, how is the information of the history compressed into the prompt?
>
> Thanks for bringing up this point. As we stated in our updated paper, the prioritised objective of this work is to provide the community a dynamic economics environments for evaluating LLMs. In the meantime, we tried to introduce the least amount of prompt engineering. Explore the optimal prompt structure for game history is beyond the scope of this work.
>
> (Continued below)

---

> ### Author Response · Authors · 2023-11-21
> **Replyt to Reviewer HCDQ**
>
> (Continued)
>
> 9. Generalisation of strategic behaviours
>
> > Can the strategic behaviors observed in LLMs within the confines of the study be generalized to other forms of competitive games or economic models?
>
> Thanks for this great question. The convergence rate we proposed in this work can be easily generalised to any economics games with unique pure NEs.

---

### Official Review · Reviewer_JmK9 · 2023-11-05

**Soundness:** 1 poor
**Presentation:** 3 good
**Contribution:** 1 poor
**Rating:** 3
**Confidence:** 3

**Summary:**

The authors consider the problem of quantifying the degree of rationality that Large Language Models can exhibit in economic settings. To this end, the authors present a set of economic games in which the authors measure the ratio of the payoff achieved by the LLMs with Nash equilibrium payoffs.

**Strengths:**

This is a really great direction, and I am very excited about the direction of this paper which seeks to bring game-theoretic tools to the study of LLMs. I hope the authors pursue these directions further!

**Weaknesses:**

The main assumption of the paper, namely that rational agents play a Nash equilibrium, is in my opinion incorrect. More generally, rational agents play a rationalizable action, i.e., the solution concept of interest is rationalizability [1]. In fact, rational agents might not be compelled to play a Nash equilibrium as summarized by following quote from Luce and Raiffa [2, page 63] regarding Nash equilibrium and rationality here is relevant:

> Even if we were tempted at first to call a (Nash) non-conformist 'irrational', we would have to admit that (his opponent) might be 'irrational' in which case it would be 'rational' for (him) to be 'irrational'-to be a (Nash) non-conformist.

The set of experimental settings are relatively standard, and do not add significantly to the existing literature.

**Questions:**

Comments and questions: The measure of rationality provided by the authors seems to become ill-defined if Nash equilibria are not unique, since the equilibrium payoffs are not unique, what do the authors do in such settings?

The definition of the rationality metric involves time, should I be thinking of this as time-step of a repeated-game?


[1] Bernheim, B. Douglas. "Rationalizable strategic behavior." Econometrica: Journal of the Econometric Society (1984): 1007-1028.

[2] Luce, R. Duncan, and Howard Raiffa. Games and decisions: Introduction and critical survey. Courier Corporation, 1989.

---

> ### Author Response · Authors · 2023-11-21
>
> We thank the reviewer for pointing out the interesting question this work tries to answer. We reply the reviewer’s concerns one-by-one below.
>
> 1. NE Rationalisability in the games
>
>  > The main assumption of the paper, namely that rational agents play a Nash equilibrium, is in my opinion incorrect. More generally, rational agents play a rationalizable action, i.e., the solution concept of interest is rationalizability [1].
>
> We thank the reviewer for the reference. In the updated Section 4.1, we mentioned that the agents’ behaviour could be a result of themselves being not maximally rational, or that they believe their opponents are not rational, thus resulting in them playing a rationalizable strategy. We attempt to disentangle the two effect by introducing a rational environment, where agents face hard-coded rational players and were informed in prompt that their opponents are rational.
>
> 2. Non-unique NEs in the games
>
> > The measure of rationality provided by the authors seems to become ill-defined if Nash equilibria are not unique, since the equilibrium payoffs are not unique, what do the authors do in such settings?
>
>
> We thank the reviewer for proposing this concern. As we stated in the updated Section 3.1, all the current games in our EconArena have only a unique pure NE. Such implementation guarantees the objectives of LLMs conflict with each other, and it is more straightforward to compare the strategies of LLMs with the NE strategy.
>
> 3. Repeated games
>
>  > The definition of the rationality metric involves time, should I be thinking of this as time-step of a repeated-game?
>
> We thank the reviewer for proposing this concern. As we stated in the updated Section 3.1, all the current games in our EconArena are of single-round to ensure that LLMs can be used in an “off-the-shelf” fashion, and no additional prompt engineering of history management is necessary. In the meantime, we keep this practice to make sure we introduce the least amount of influence from prompt engineering onto the performance of LLMs in EconArena.

---

### Author Response · Authors · 2023-11-21
**Overall Response**

We thank all the reviews for identifying the interesting direction proposed in this work, and the valuable feedback about updating the paper. We made many major updates to the previous draft, and they are summarised as follows:

1. Title

We changed the title to “Economics Arena for Large Language Models” to emphasise more on the economics arena (EconArena for short) introduced by this work. EconArena is a dynamic environment consists of various types of economics games, and is specifically designed for interacting with LLMs. It will be released as both a Python module available on PyPI, and on a website where games can be run through GUI operations.

2. Methodology

We rewrote the methodology section to emphasise the deign principles of our economics arena, as well as the potential metrics of interest from either economics perspective or computer science perspective.

3. Experiment about the rational behaviours of the LLMs-based agents

We introduced a rational environment, where agents play against hard-coded rational agents and were explicitly informed that they are playing with rational players.

4. Experiment about the adaption of LLMs to dynamic environments

We introduced variations in game configurations, opponent types and prompt languages into experiment settings to demonstrate LLMs’ adaptive behaviour when facing dynamic environments.

5. Experiment about the strategic reasoning ability of LLMs

We updated experiments where historical information are revealed to LLMs. The adjustment in strategies when given information about past plays demonstrate LLMs’ strategic reasoning ability as they take into consideration their opponents’ strategies.

6. Architecture and prompt details

We introduced the architecture of EconArena and provided the prompts used in our experiments in Appendix A and B respectively.


We apologised for updating very much of the previous draft, and introduced many new contents during the rebuttal period.

Thanks again for the time and efforts of all reviewers.

---

### Meta-Review · Area_Chair_BGCU · 2023-12-06

**Metareview:**

The paper studies an interesting topic, i.e., the strategic reasoning ability of LLMs by letting the play in competitive games. This is a very interesting question. The AC appreciated and acknowledges the effort the authors put during the rebuttal to improve their manuscript based on the reviewers comments. However it seems that the reviewers (unanimously) were not convinced that the paper is above the bar for ICLR due to the limitations of the experiments.

**Justification For Why Not Higher Score:**

The reviewers unanimously believe that the paper is below the bar and that further revisions are needed (with more experiments).

**Justification For Why Not Lower Score:**

Not applicable

---

### Decision · Program_Chairs · 2024-01-16

Reject